# The Effect of Plasma Treatment of Polyethylene Powder and Glass Fibers on Selected Properties of Their Composites Prepared via Rotational Molding

**DOI:** 10.3390/polym14132592

**Published:** 2022-06-26

**Authors:** Zoya Ghanem, Hana Jelinek Šourkova, Jan Sezemsky, Petr Špatenka

**Affiliations:** Department of Materials Engineering, Faculty of Mechanical Engineering, Czech Technical University in Prague, 12135 Prague, Czech Republic; hana.sourkova@fs.cvut.cz (H.J.Š.); jan.sezemsky@fs.cvut.cz (J.S.); petr.spatenka@fs.cvut.cz (P.Š.)

**Keywords:** rotational molding, polyethylene, glass fiber, composites, plasma treatment, adhesion

## Abstract

In this article, the effect of plasma treatment of polyethylene powder and glass fibers on the adhesion between polyethylene and glass fibers in composites prepared by rotational molding was studied. In contrast to other processing techniques, such as injection molding, the rotational molding process operates at atmospheric pressure, and no pressure is added to ensure mechanical interlocking. This makes reinforcing the rotomolded product very difficult. Therefore, the formation of chemical bonds is necessary for strong adhesion. Different combinations of untreated polyethylene (UT.PE), plasma-treated polyethylene (PT.PE), untreated and plasma-treated glass fibers were manually mixed and processed by rotational molding. The resulting composites were cut and tested to demonstrate the effect of the treatment on the adhesion between the composite components and on the mechanical properties of the final composites. The results showed that the treatment of both powder and fiber improved the adhesion between the matrix and fibers, thus improving the mechanical properties of the resulting composites compared to those of pure polyethylene samples and composites prepared using untreated components. The tensile strength, tensile modulus, and flexural modulus of the composites prepared using 10-min treated powder with 20 wt% of 40-min treated fibers improved by 20%, 82%, and 98%, respectively, compared to the pure polyethylene samples.

## 1. Introduction

Rotational molding is a molding technique used to produce hollow plastic parts with equal wall thickness and very low residual stresses in the final products.

The process is used to produce a wide range of predominately hollow products of different sizes and shapes including storage tanks, shipping containers, kayaks, and barriers [1,2]. Manipulation of parameters such as the reduction of cycle time [3,4,5] and bubble removal [6,7] improved the properties of rotomolded samples to a greater extent [8].

However, the main disadvantages of this process are high temperature and long production cycle requirement, which limit the materials that can be constructed into a specific type of polymers capable of withstanding elevated temperatures for a relatively long period. These materials which are dominated by different grades of polyethylene are considered unsuitable in applications where product strength and rigidity are important. Therefore, the production of composite materials with improved mechanical properties and thermal stability by rotational molding has attracted the attention of researchers over the past few decades. 

There have been many attempts to incorporate different fiber types into the rotational molding process. The first attempt to produce composites via rotational molding was conducted by Torres et al. in 2003 [9]. In this study jute, wood, cabuya, pecan, sisal fibers and different types of rice shell flour were used as a filler for high-density polyethylene. Following this, many studies investigating the possibility of producing composites via rotational molding were conducted. Different natural fibers such as flax [10], banana and abaca [11], agave [12,13,14], coir [12,14,15], pine [12], maple [16,17], wood [18,19], buckwheat husk [20], glass fibers [21,22], carbon fibers [22], glass particles [23,24], other synthetic particles [24,25] and nanoparticles [26,27,28,29,30] as reinforcements with different grades of polyethylene [9,11,12,14,15,17,21,22,31], polyamide [26], polylactic acid [13,20], and polypropylene [18] as matrix were used in the studies.

However, no research has yet been reported using it successfully in industry. Contrary to other processing techniques such as injection molding and pressure molding, the rotational molding process operates at atmospheric pressure which makes reinforcing of the rotomolded product very difficult. The main problems that arise while using reinforcements in rotational molding are nonuniform distribution of the filler inside the matrix and poor adhesion between the fillers and the matrix, as no pressure is added to ensure mechanical adhesion between the two phases. This leads to the segregation and the agglomeration of the reinforcements. To overcome these problems, researchers have tried different chemical surface treatments of the fillers and/or the addition of coupling agent to the polymer matrix. Wang et al. studied the effect of three different chemical treatments of flax fibers on the mechanical properties of LLDPE/flax fiber composites prepared via rotational molding. Silane, benzoylation, and peroxide were used for treatment; they reported that silane treatment was the best for improving mechanical properties and water absorption rate [10]. Ortega et al., prepared two- and three-layered banana and abaca fibers/polyethylene composites via rotational molding, and studied the effect of NaOH treatment of the fibers on the resulted composites. The results of the mechanical tests showed that the addition of banana and abaca fibers improved the tensile and flexural modulus, while the tensile and impact strength decreased, and NaOH treatment improved the properties of the composites [11]. Lopez et al. used agave, coir, and pine as reinforcements for LLDPE, with and without MAPE (maleic anhydride grafted polyethylene). The morphological tests showed that surface treatment helped to achieve better adhesion between the fibers and the polymer in all cases. In addition, mechanical properties of the composites with 20 wt% treated fibers were higher than the mechanical properties of net polyethylene and of composites prepared using untreated fibers [13]. Hanana et al., tried maple fibers as reinforcement for the LLDPE matrix and produced a composite using rotational molding. Maple fibers were treated with malleated polyethylene (MAPE) and properties of the composites were compared to composites prepared with untreated fibers. Tensile and flexural modulus increased for both composites prepared using treated and untreated fibers compared to the net samples. However, in all cases composites with treated fibers exhibited higher values compared to composites with untreated fibers. The tensile strength of the untreated fibers composites decreased with increasing fiber content compared to unfilled samples, while the tensile strength of the treated-fiber composites was slightly above the unfilled samples. Impact strength decreased with increased fiber content for all composites, with slightly higher values for treated-fiber composites compared with untreated-fiber composites [17]. 

Cold plasma treatment for adhesion improvement has attracted the attention of researchers as an effective and environmentally friendly alternative of chemical treatment. The treatment can be used to treat both matrix and fibers to improve their wettability and draft different functional groups on their surfaces. The type of functional group depends on plasma composition, and predominately carboxyl, hydroxyl, amine, or aldehyde groups are investigated as a tool for improvement of filler–matrix interfacial properties [32]. Plasma treatment of glass fibers improved their adhesion to both polyester and epoxy resins [33,34]. J. Trejbal et al. reported that oxygen plasma treatment of glass fibers increased their wettability by 25% when compared with untreated glass fibers [35]. Plasma treatment of wood fibers improved the tensile strength and modulus of polypropylene composites prepared using the treated fibers when compared with composites prepared using untreated fibers [30]. Similarly, plasma-treated coir fibers/starch composites [36] and plasma-treated flax fibers/LDPE composites [37] had better mechanical properties when compared with composites prepared using untreated fibers. Another study reported that plasma treatment of carbon successfully improved their adhesion to different matrixes [38,39,40]. 

In rotational molding, Rodrguez et al. used a mixture of polyethylene with untreated and plasma-treated carbon nanofibers to prepare nanocomposites with different fiber content (0.01, 0.1, and 1.0 wt%). SEM images showed that the treated carbon fiber had a better distribution inside the matrix as a result of good adhesion and fiber wetting. Mechanical testing showed that plasma-treated nanofibers increased the impact strength of composites compared to net polyethylene and composites prepared by untreated nanofibers. The tensile modulus and tensile strength also increased by 20% and 8%, respectively. S. Panikkassery Sasidharan et al. studied rotomolded composites prepared with untreated polyethylene (PE) and plasma-modified polyethylene (PPE) and 5 wt% of untreated and alkali-treated coir fiber. The results showed that the composites prepared using treated matrix and treated fibers had the lowest water absorption rate and the studied mechanical properties of these composites improved compared to untreated materials [15].

In this article, the effect of plasma treatment of both LLDPE and glass fibers on tensile strength and modulus, flexural modulus and impact strength of their composites prepared by rotational molding were studied. In addition, the effect of the treatment on adhesion between the composite’s components was also investigated. 

## 2. Materials and Methods

### 2.1. Materials

Linear low-density polyethylene (LLDPE) ‘DOWLEX 2629UE’ was used as matrix, with density 935 kg/m^3^ and melt flow index (MFI) (190 °C/2.16 kg) 0.004 kg/10 min from Dowlex Chemical Company, (Midland, MI, US). Plasma modification of PE powder was carried out by Surface Treat, a.s. (Turnov, Czech Republic). 

Short, milled glass fibers with an average length of 0.19 mm and an average diameter of 14 μm by LanXESS Company (Cologne, Germany) were used as reinforcement. 

The release agent was Rotorelease® MKX-17-014 from Münch Chemie International GmbH (Weinheim, Germany).

### 2.2. Plasma Treatment 

Plasma treatment of the powder and the fibers was conducted in a laboratory device LA 400 (Surface Treat, a.s., Turnov, Czech Republic), with a vacuum working chamber and microwave discharge (1 kW, pulse mode, 0.25 kg of powder), pressure 100 Pa, oxygen was used as working gas with flow 100 sccm, powder was treated for 1 min, 3 min, 5 min and 10 min, while glass fibers were treated for 20, 40 and 60 min. A detailed description of the plasma reactor and plasma-treatment process is provided in the previous publications of H. Sourkova and P. Spatenka [41].

### 2.3. Samples Preparation

The samples were prepared using a laboratory-scale ‘rock and roll’ rotational molding machine with electrical heating (Figure 1a). The machine is designed to undergo full 360° rotation motion of the mold around one axis (rolling) and swinging action of the oven around a perpendicular axis (rocking). The oven is heated using four electric resistance heaters placed at the bottom of the oven. Cooling is carried out using a fan fixed under the oven. The temperature inside the oven and the molds are monitored using two sensors, one is fixed in the oven to measure the oven temperature, and the other is inserted inside the mold after fixing the mold in the oven, which measures the internal air temperature of the mold (PIAT). The rotation speed was fixed at 10 rmp for all experiments, and the swing angle was 45°. The mold was a rectangular aluminum box with dimensions 250 mm × 95 mm × 95 mm. The shape of the mold and of the rotomolded samples are presented in Figure 1b. Before loading the material into the mold, the release agent was applied to the inner surface of the mold.

The weight of the material used to produce each sample was 0.3 kg. The material used was pure PE powder or PE powder/glass fiber mixture to produce samples with wall thicknesses between 3 and 4 mm. The powder and fibers were mixed before molding for 5 to 10 min using a kitchen mixer to ensure proper distribution of the fibers in the matrix.

In the first stage of the research, the effect of plasma treatment of the polyethylene powder was studied. Polyethylene powder was treated for 1, 3, 5, and 10 min. The selected treatment times were chosen according to the findings of H. Sourkova and P. Spatenka [41]. They demonstrated that a significant increase in wettability can be observed at up to 10 min of treatment time, while concentration of the oxygen groups increases with increasing treatment time. The treated powder was mixed with 10 wt% untreated glass fibers and then used to prepare the composites. The material was heated to an oven temperature of 250 °C and held at this temperature until the PIAT temperature reached 220 °C, at which point the heating was stopped, and cooling process started.

In the second stage, we studied the effect of the heating process on the final properties of the composites. Two different temperatures were tested. Instead of using the PIAT temperature to determine when to stop heating and start cooling, in this stage the oven was heated to 250 °C or 220 °C and held at these temperatures for different periods and at the end the of the holding period, the heating was stopped, and the cooling process was started. The hold time at the oven max temperature was 15 min, 30 min, 45 min and 60 min. Composites were prepared using 5-min treated powder and 10 wt% untreated fibers.

In the third stage, the effect of plasma treatment of glass fibers was studied. Glass fibers were treated for 20, 40 and 60 min and then mixed with untreated powder and 10-min plasma-treated powder. The composites were prepared at oven temperature 220 °C and hold time 30 min.

Finally, the effect of different fiber content was studied by preparing composites using 10-min treated powder mixed with 10, 15 and 20 wt% of 40-min treated glass fibers. The oven temperature was 220 °C with a hold time of 30 min.

### 2.4. Testing Methods

Tensile strength and modulus were measured according to ASTMD638 using the TINUS OLSEN H50KT universal testing machine (Tinius Olsen, Ltd., Salfords Surrey, UK) at a gauge length of 60 mm and a speed of 50 mm per minute at room temperature. The three-point bending test was performed according to ASTM D790 using MTS Exceed E42 bending machine (MTS Systems, Eden prairie, MN, USA), and the testing speed was 10 mm/min. Charpy impact tests were performed using CEAST 7.5 J (Instron, Norwood, MA, USA), according to ASTM D6110. The samples were notched with a 2 mm offset. The tests were performed at room temperature. 

Scanning electron microscopy (SEM) images were captured using a scanning electron microscope (Lyra3, Tescan) (Tesacan, Brno, Czech Republic) and optical microscopy images (OM) images were obtained using a Nikon SMZ 1500 stereomicroscope (Nikon, Tokyo, Japan) equipped with a CCD camera.

The reported values of all the tests are the averages of at least five specimens. To obtain the specimens, we first prepared the rotomolded samples via rotational molding for each combination in each stage. The rotomolded boxes were cut into 6 sheets (4 side walls and upper and lower walls) using an electrical cutting machine. Specimens for each test method with suitable shape and dimension were then cut from the different side walls of different rotomolded samples using laser cutting machine Snapmaker A250T (Snapmaker, Shenzhen, China).

## 3. Results and Discussion

### 3.1. The Effect of Different Factors on the Selected Mechanical Properties of the Composites

#### 3.1.1. Effect of Plasma Treatment of Polyethylene Powder

The tensile strength of the composites prepared using powder treated for different lengths of time is presented in Figure 2. The tensile strength of the composites prepared using untreated powder maintained almost the same values as the net polyethylene samples. Increasing the plasma treatment time of the powder slightly increased the tensile strength. Composites prepared using 10-min treated powder had a tensile strength 7% higher than the tensile strength of pure polyethylene and 9% higher than composites prepared using untreated powder.

The reason that significant improvement in the tensile strength was not achieved as a result of different treatment times could be that at short treatment times (1 and 3 min) the treatment was not yet sufficient to improve the adhesion between the fiber and the matrix, while at long treatment times (5 and 10 min) the reason could be an increased number of bubbles on the outer surface of the samples and uneven inner surface indicating agglomeration of the fiber. 

The tensile modulus is presented in Figure 3. All composites showed a higher modulus than pure polyethylene, the tensile modulus of the composites increased from 272.2 MPa for composites prepared with untreated powder to 343.8 MPa for composites prepared using 3-min treated powder. The reason for this increase is the presence of rigid fibers in the matrix. However, the modulus decreased when using 5-min and 10-min treated powder; this decrease resulted from the higher bubble content in the composites and uneven inner surface.

The flexural modulus presented in Figure 4 showed a similar trend to the tensile modulus. The composites showed a higher modulus than the net polyethylene in all cases, and the highest modulus was reached when 1-min treated powder was used to prepare the composites which was 47% higher compared to the modulus of net polyethylene. This is also similar to the tensile modulus where a slight decline can be noticed in the modulus when using longer time treated powder, but this decline appeared earlier in the case of flexural strength where a 10% decrease in the modulus can be noticed when using 3-min treated powder compared to composites prepared using 1-min treated powder.

W. Chang et al. [21] and Höfler et al. [22], also reported the improvement of both tensile and flexural modulus of the samples prepared via rotational molding as a result of incorporation of different types of glass fibers to the polyethylene compared to unfilled samples. S. Panikkassery Sasidharan et al. [15] reported the improvement of tensile modulus of coir fiber/plasma treated polyethylene composites prepared via rotational molding comparing with unfilled samples and with composites prepared using untreated powder. Improvements in tensile modulus and flexural modulus of other natural fiber composites prepared by rotational molding compared to unfilled samples were also reported in other studies [12,14,17]. 

The impact strength of the composites is presented in Figure 5. All composites showed lower impact strength than those of unfilled samples; this is a result of the presence of hard phase in the mixture. Increasing the treatment time also contributed to a further reduction of impact strength, which as mentioned previously, is a result of a higher number of bubbles on the surface of composites prepared using powder treated for more than 1 min. The highest reduction in impact strength was observed for composites prepared using 5-min treated powder where the reduction reached 21% compared to unfilled samples, while a recovery can be noticed when 10 min powder was used. However, the value of the impact strength was still less than that of net polyethylene. Similar reductions in impact strength were noticed in references as a result of the use of glass fibers for reinforcement in rotational molding [21,22]. Contrary to the glass fibers, one of the natural fibers composites showed to be more effective in improving the impact strength of the composites prepared via rotational molding [14,15].

OM images of the outer surface of samples prepared using different treated powders are presented in Figure 6. An increase in bubble numbers can be noticed with increasing treatment time.

This behavior can be explained by the fact that the powders treated for a longer time need longer sintering times to achieve a uniform and proper even layer [42]. In the first step of the experiment, the holding time at the maximum oven temperature 250 °C was only 5 min, and this could not be sufficient for proper sintering of the mixture of glass fiber and powder treated for a longer time.

#### 3.1.2. Effect of Oven Temperature and Holding Time

The tensile strength of the composites prepared at different oven temperatures and different holding times are presented in Figure 7.

Preparation of the composites at an oven temperature of 250 °C for 15-min holding time improved the tensile strength by 9% compared with composites prepared at an oven temperature of 250 °C and PIAT 220 °C (holding time in this case was only 5 min). However, 30-min holding time at 250 °C decreased the tensile strength again to 18.4 MPa and the samples already showed degradation signs. This is why no longer holding times were tested at this temperature, instead a lower temperature of 220 °C was tested for different holding times. Composites prepared at 220 °C for 15 min had the lowest tensile strength, and visual inspection showed an abundance of bubbles on the outer surface and an uneven inner surface. Increasing the holding time to 30 min at the same temperature increased the tensile strength to a value similar to that of the composites prepared at a temperature of 250 °C with a holding time of 15 min. Further increases of holding time did not cause any further improvement in tensile strength. The improvement of tensile strength as a result of longer holding time could be explained by better sintering of the samples. No bubbles were noticed on the outer surface, and the inner surface was smooth and even, which indicates better sintering of the powder and better distribution of the fibers in the matrix.

The effect of different oven temperature and holding time on the tensile modulus is presented in Figure 8. The highest tensile modulus value was obtained for the composite prepared at an oven temperature of 220 °C for 30 min, and it was 16% higher compared to our original composites prepared at oven temperature 250 °C with 5-min holding time. This resulted from less bubbles and smoother inner surface.

Similarly, it was found that the flexural modulus of composites prepared at an oven temperature of 220 °C for a holding time of 30 min was the best, which is 26% better than composites prepared at an oven temperature of 250 °C and holding time of 5 min (Figure 9).

The effect of different temperatures and holding times on the impact strength of the composites is presented in Figure 10. A different temperature and longer holding time at the peak temperature of the oven helped to increase the impact strength slightly as a result of decreasing the number of bubbles and a smoother inner surface, but the impact strength in all cases stayed less than the impact strength of unfilled samples.

Figure 11 shows the optical microscopy images of the outer surface of samples prepared at oven temperature 220 °C for 30 min, and no bubbles can be noticed. This indicates that a longer heating time is needed to prepare composites using powder treated for a longer time. These results correspond with the results obtained by Z. Weberova et al., on the sintering of plasma-treated polyethylene on glass rods in an oven which showed that plasma treatment time of the polyethylene powder affected their sintering behavior and longer time is needed to obtain a proper sintering of the powder [42]. 

#### 3.1.3. Effect of Plasma Treatment of Glass Fibers

The tensile strength of the composites prepared using treated and untreated glass fibers is presented in Figure 12. Tensile strength of composites prepared using a mixture of untreated powder with plasma treated fibers had almost the same tensile strength as the composites prepared using untreated powder and untreated fiber. On the other hand, composites prepared using mixture of treated powder and treated fiber had higher tensile strength values compared with composites prepared using untreated fiber. Using both treated powder and treated fibers increased the tensile strength to 21.8 MPa for composites prepared using 10-min treated powder and 40-min treated fibers, which is 17% higher than unfilled PE and 15% higher than composites prepared using untreated powder and fibers. The reason for this increase could be that the treatment of glass fibers added oxygen groups to the fiber surfaces and increases in their surface energy with respect to the surface energy of the powder, increasing their wettability, and hence the adhesion between the matrix and the fibers. As is well known, for optimal wetting of any substrate by a liquid, the surface energy of the liquid should be higher than the surface energy of the substrate. Plasma treatment of polyethylene powder increases not only the concentration of the active groups on the powder surface but also the surface energy of the powder to values which could become close or even higher to the glass fibers surface energy, which could decrease the wettability. Therefore, treatment of the fibers was necessary to ensure that their surface energy was higher in relation to the powder surface energy. A. Haji et al., reported improvement of the tensile strength of epoxy/glass fibers composites as a result of the plasma treatment of the glass fibers when compared with composites prepared from untreated fibers [34]. 

Plasma treatment of the fibers did not significantly affect the tensile modulus (Figure 13). Composites prepared using untreated powder with treated glass fibers had almost the same values as composites prepared using untreated powder and untreated glass fibers. A very small decline less than 5% can be observed for composites prepared using untreated powder with 20-min treated fibers. Similarly, a minor decline of all composites prepared using treated powder and fibers was noticed when compared with composites prepared with untreated fibers.

Figure 14 presents the flexural modulus of composites prepared using treated glass fiber. The flexural modulus of composites prepared using untreated powder and 20-min treated fiber improved by around 10% compared to composites prepared by untreated powder with untreated fibers; no further improvement was observed as a result of using treated glass fibers treated for 40 and 60 min. The composites prepared with treated powder and treated fibers maintained almost the same values as the composites prepared using untreated components, regardless of the fiber treatment time. 

Figure 15 presents the effect of plasma treatment of glass fibers on the impact strength of composites. Composites with treated powder and treated fibers maintained the same impact strength compared to those of untreated components. Similar behavior was also noticed for composites prepared with both treated powder and glass fibers; the only exception was a minor decline by 7% noticed for composites containing 20-min treated fibers. Fiber treatment did not help to increase the impact strength to values higher than net polyethylene values.

#### 3.1.4. Effect of Glass Fibers Content

The effect of fiber content on the tensile strength of the composite is presented in Figure 16. The composites were prepared using 10-min treated powder and 40-min treated glass fibers at oven temperature 220 °C and 30-min holding time. The tensile strength of composites increased by increasing fiber content up to 20 percent. Tensile strength of composites containing 20 wt% fibers increased by 20% compared to net polyethylene. This resulted from optimizing the process conditions and using both treated powder and treated glass fibers, which improved the adhesion between the fibers and the matrix.

The effects of the fiber content on the tensile and flexural modulus of composites prepared using 10-min treated powder and 40-min treated fibers are presented in Figure 17 and Figure 18, respectively. Both moduli increased with the addition of a higher content of fibers, which was the result of the incorporation of a greater number of rigid fibers in the matrix. The tensile modulus increased by 82% for composites containing 20% fibers, while flexural modulus increased by 99% of the same composites compared to unfilled PE samples. 

The results of a higher content of glass fiber on the impact strength is presented in Figure 19. As shown, the impact strength kept decreasing with increasing fiber content to reach 38% at 20 wt% glass fiber content when compared with unfilled samples. This could be a result of the presence of bigger amount of rigid phase, which hinders the movement of polymer chains and decreases the total ability to absorb and transfer impact energy.

#### 3.1.5. The Effect of Plasma Treatment of the Powder and the Fibers on the Stress-Strain Behavior of the Composites

Stress–Strain diagram obtained from the tensile test is presented in Figure 20. The stress–strain diagram of composites prepared from untreated powder and fibers was identical to that of the unfilled samples diagram, while treatment of the powder for 10 min changed the behavior of the composites. As can be noticed, the composites prepared using the treated powder endured higher stress and showed less deformation under the stress compared to unfilled samples and to the composites prepared with untreated components. This indicated that the plasma treatment was successful in creating adhesion between the powder and the fibers, leading to efficient reinforcement of the polyethylene by the fibers and subsequently better tensile behavior of the composites. Further improvement in the tensile behavior can be noticed as a result of the treatment of the glass fibers and the increase of fiber content to 20%.

Figure 21 presents stress–strain diagram of flexural properties. Similar to tensile behavior, incorporation of fibers to the polyethylene changed its bending behavior. The composites tolerated higher stress with less deformation compared to unfilled samples. The plasma treatment and the higher fibers content also had a positive effect on the values of maximum stresses.

### 3.2. Morphology

SEM images of fractured surfaces of composites prepared using powder treated for different times are presented in Figure 20. The surface of composites prepared using the untreated power (Figure 22a) shows no adhesion between the fibers and the matrix; a gap between them can be clearly seen. For composites prepared with 1-min treated powder (Figure 22b), the gap between the matrix and the fibers can still be seen and no trace of polyethylene is shown on the fiber surface. With increasing of the treatment time to 3 min (Figure 22c), the gap disappeared, and little trace of polyethylene can be seen on the fiber surface. Evidence of even better adhesion can be seen in Figure 22d,e, which presents the fracture surface of samples prepared using 5-min and 10-min treated powders, respectively. The images show that there are no gaps between the fibers and PE matrix, and there is an increase in the amount of polyethylene residue left on the fiber surface after breaking; this indicates good adhesion between the fiber and the matrix. These images are consistent with those of Zuzana et al. [42]. on the adhesion between treated material and glass rods.

SEM images taken from different locations on the fracture surface of the composites prepared using 10-min treated powder with 40-min treated fibers are presented in Figure 23. Improved adhesion between the powder and the fibers as a result of the use of treated powder along with treated fibers can be clearly seen. Figure 23c shows a broken fiber completely covered with polyethylene that indicates a very good adhesion between the fiber and the matrix. Other broken fibers can be observed in Figure 23a,b. This could be due to a higher concertation of the functional group on the surface of treated glass fibers and treated polyethylene powder, and a higher surface energy of glass fibers after treatment. Similarly, good adhesion was not achieved in the literature where short glass fibers were dry mixed with the powder and used as reinforcements for composites prepared by rotational molding [21,22]. Only an improvement in the adhesion was noticed when the fibers and the powder were pre-compound prior to molding [21].

SEM images with higher magnification of the interface between the fiber and the matrix are presented in Figure 24. Figure 24a shows the interface between untreated powder and untreated glass fiber. A gap can be clearly seen between the fiber and the matrix due to poor adhesion between two nonpolar surfaces. Moreover, the fiber surface is totally clean, and no polyethylene is left. Figure 24b shows the interface between 10-min plasma-treated powder and 40-min plasma-treated fiber, it is clearly observed that the fiber surface is covered by polymer matrix, which indicates that treatment of both powder and glass fibers was necessary to achieve sufficient adhesion between the fibers and the matrix.

## 4. Conclusions

Plasma treatment of polyethylene and glass fibers proved to be a successful method for improving the adhesion between the matrix and the fibers in composites prepared via rotational molding. The results showed that treatment of both powder and fibers is necessary to achieve optimum adhesion and improve the tested mechanical properties of the composites. Different treatment times for both powder and fibers were tested, and the results indicated it is necessary to treat the polyethylene powder for more than 5 min to improve adhesion. However, increasing the powder treatment time affected its sintering behavior and a longer heating time was needed to obtain good composites. The optimum composites were produced at an oven temperature of 220 °C for 30 min holding time, using a mixture of 10-minute treated powder and 40-minute treated fiber. The tensile strength, tensile modulus, and flexural modulus of these composites were improved by 20%, 82% and 98%, respectively, compared to pure polyethylene samples.

## Figures and Tables

**Figure 1 polymers-14-02592-f001:**
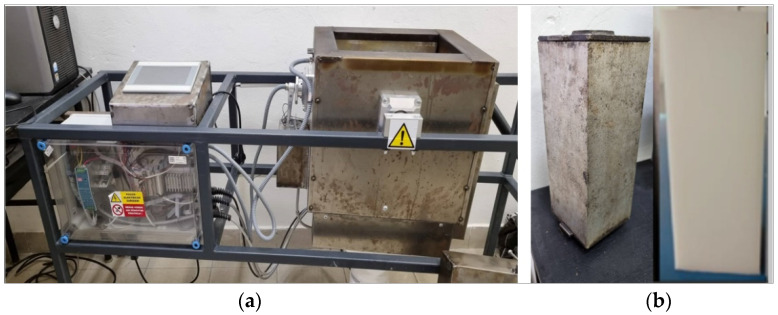
(**a**) Rotational molding machine, (**b**) The shape of the mold and samples.

**Figure 2 polymers-14-02592-f002:**
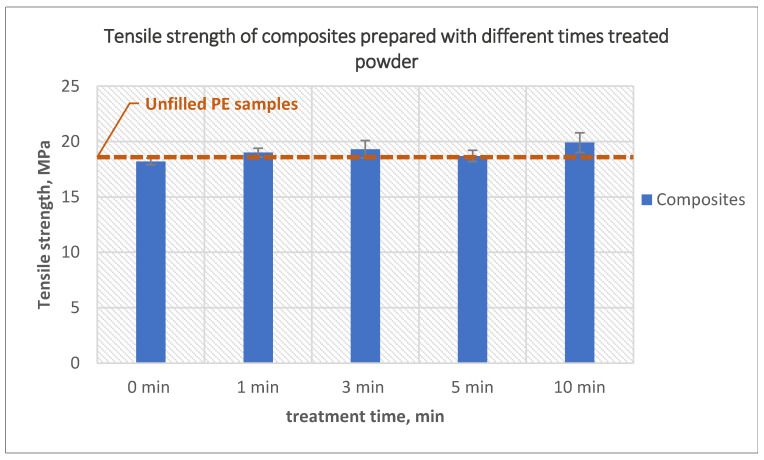
The effect of PE powder treatment time on the tensile strength of the composites produced using treated powder and 10 wt% untreated glass fibers.

**Figure 3 polymers-14-02592-f003:**
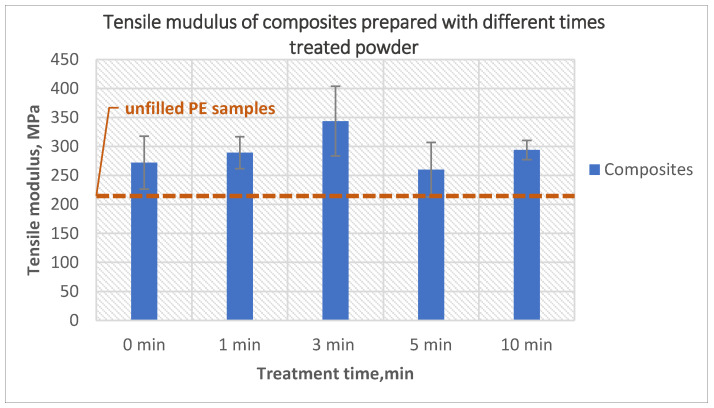
The effect of the PE powder treatment time on the tensile modulus of the composites produced using treated powder and 10 wt% untreated glass fibers.

**Figure 4 polymers-14-02592-f004:**
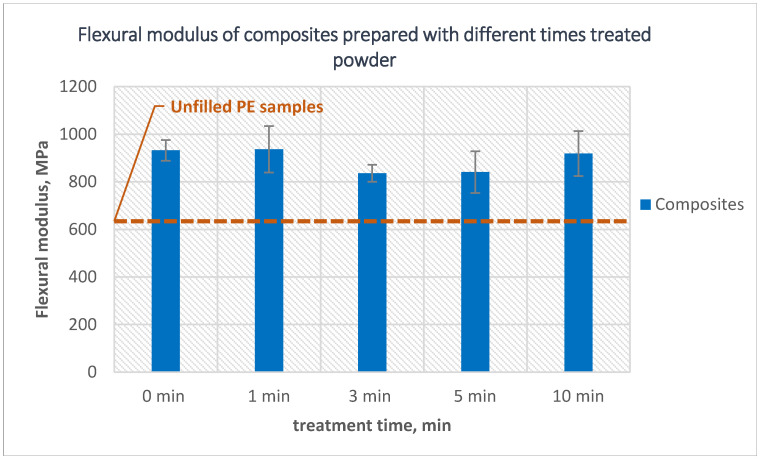
The effect of the PE powder treatment time on the flexural modulus of the composites produced using treated powder and 10 wt% untreated glass fibers.

**Figure 5 polymers-14-02592-f005:**
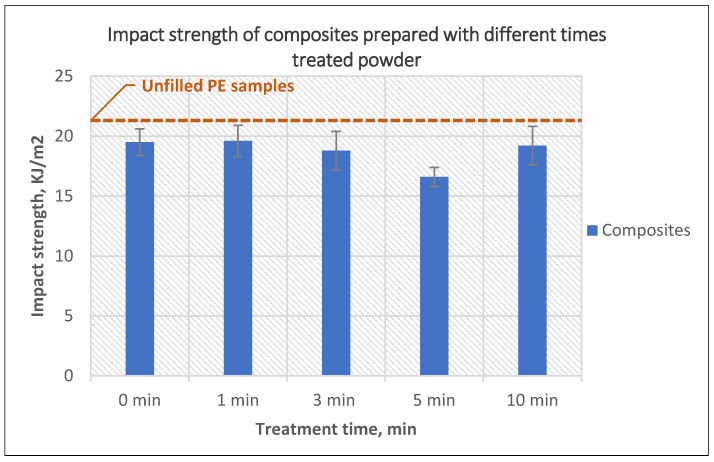
The effect of the PE powder treatment time on the impact strength of the composites produced using treated powder and 10 wt% untreated glass fibers.

**Figure 6 polymers-14-02592-f006:**
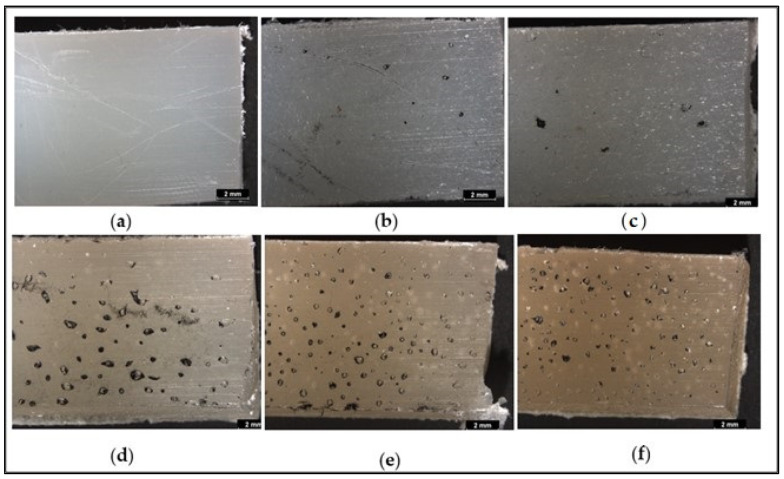
Optical microscopy images of the outer surface of samples prepared using untreated and plasma-treated polyethylene with 10 wt% untreated glass fibers: (**a**) untreated PE without fiber, (**b**) composites with untreated PE, (**c**) composites with 1-min treated PE, (**d**) composites with 3-min treated PE, (**e**) composites with 5-min treated PE, (**f**) composites with 10-min treated PE.

**Figure 7 polymers-14-02592-f007:**
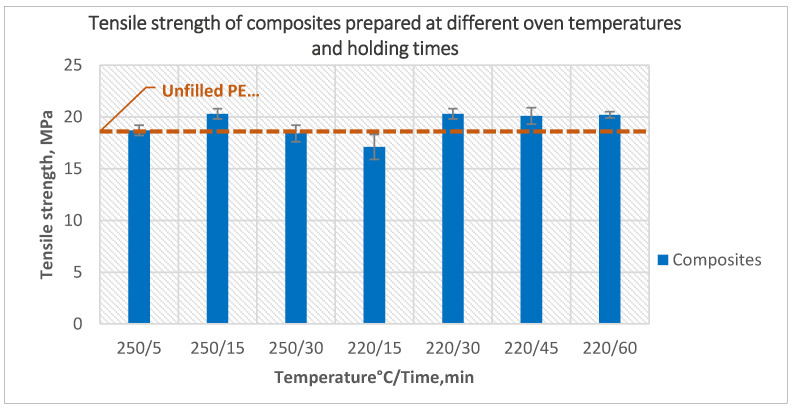
The effect of oven temperature and holding time on the tensile strength of the composites prepared using 5-min treated powder and 10 wt% untreated fibers.

**Figure 8 polymers-14-02592-f008:**
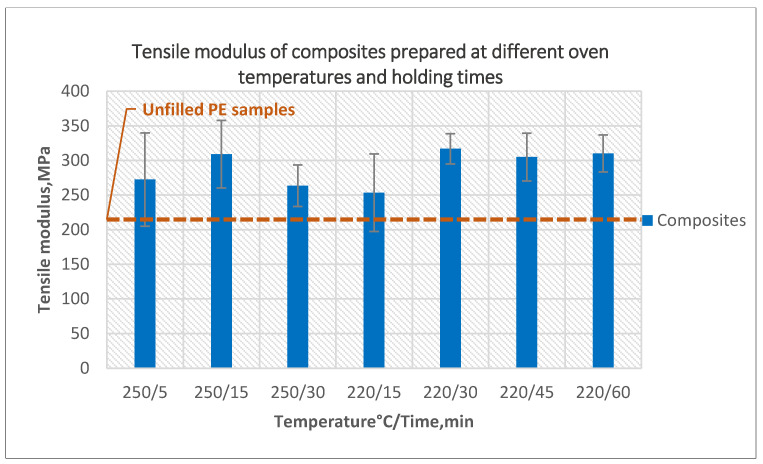
The effect of oven temperature and holding time on the tensile modulus of the composites prepared using 5-min treated powder and 10 wt% untreated fibers.

**Figure 9 polymers-14-02592-f009:**
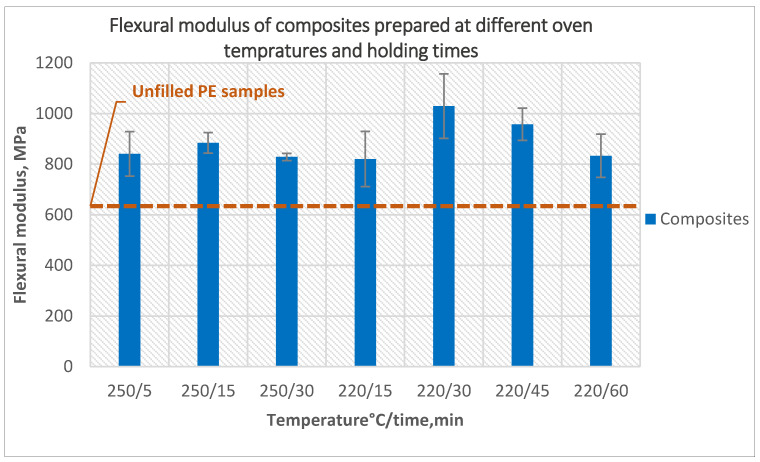
The effect of oven temperature and holding time on the flexural modulus of the composites prepared using 5-min treated powder and 10 wt% untreated fibers.

**Figure 10 polymers-14-02592-f010:**
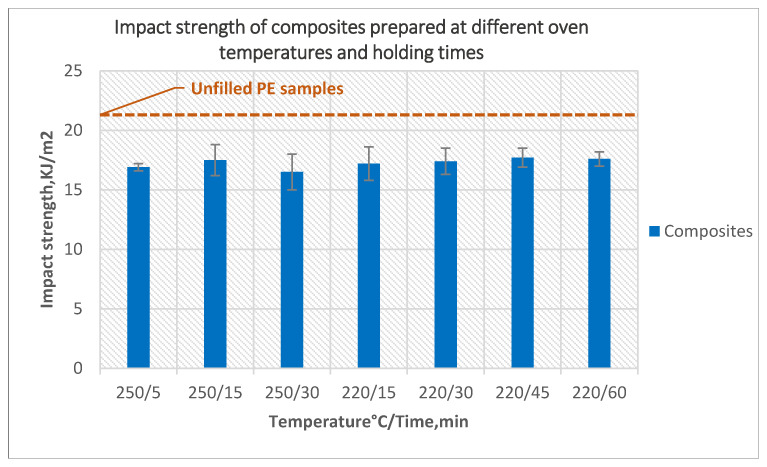
The effect of oven temperature and holding time on the impact strength of the composites prepared using 5-min treated powder and 10 wt% untreated fibers.

**Figure 11 polymers-14-02592-f011:**
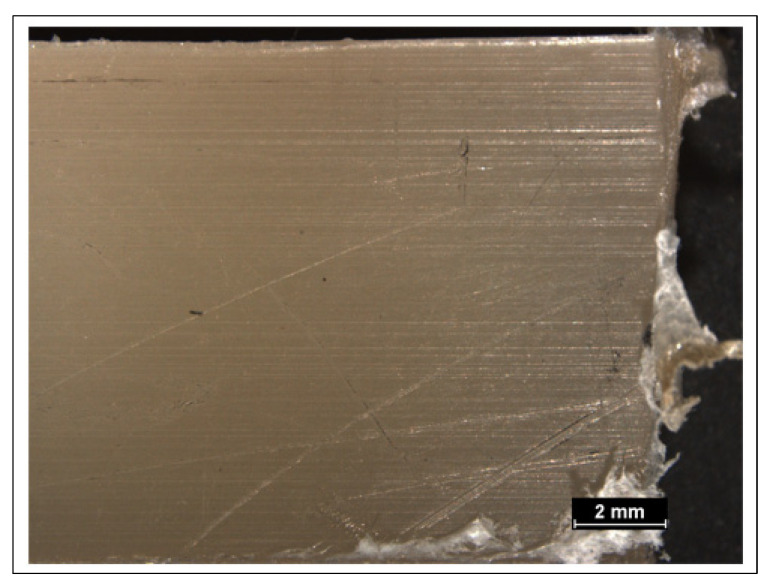
Optical image of the outer surface of composites prepared at oven temperature 220 °C and 30-min holding time.

**Figure 12 polymers-14-02592-f012:**
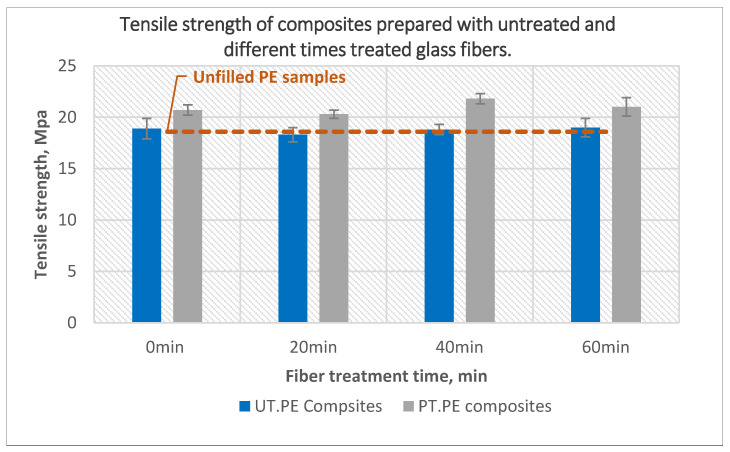
The effect of glass fiber treatment time on the tensile strength of the composites prepared using untreated and 40-min treated PE with 10 wt% untreated and plasma-treated glass fibers.

**Figure 13 polymers-14-02592-f013:**
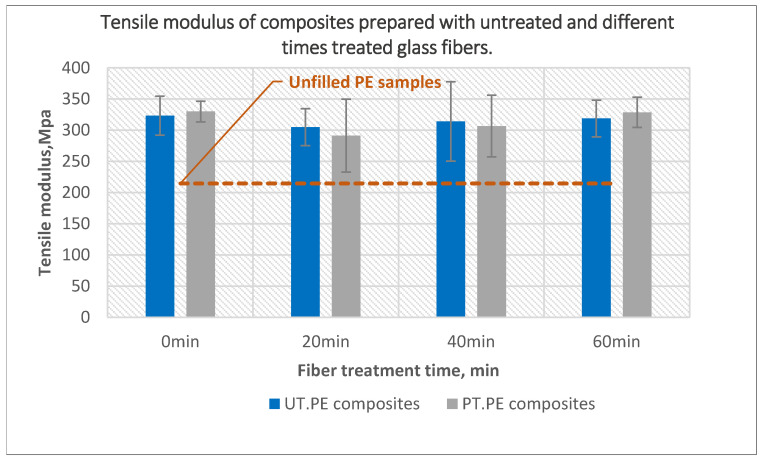
The effect of glass fiber treatment time on the tensile modulus of the composites prepared using untreated and 40-min treated PE with 10 wt% untreated and plasma-treated glass fibers.

**Figure 14 polymers-14-02592-f014:**
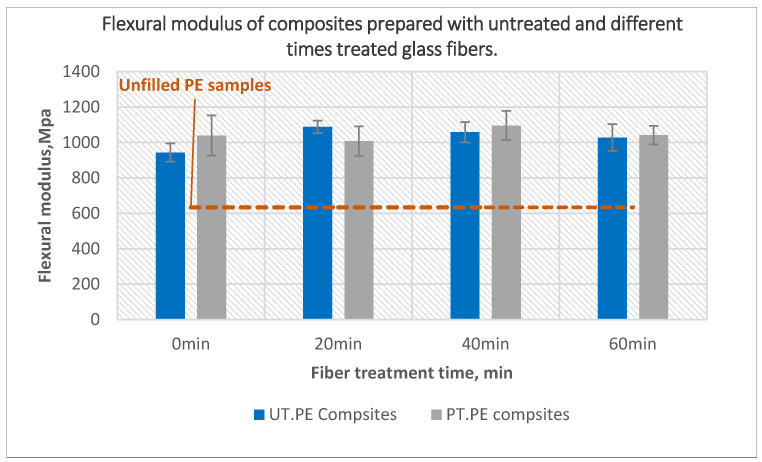
The effect of glass fiber treatment time on the flexural modulus of the composites prepared using untreated and 40 min treated PE with 10 wt% of untreated and plasma-treated glass fibers.

**Figure 15 polymers-14-02592-f015:**
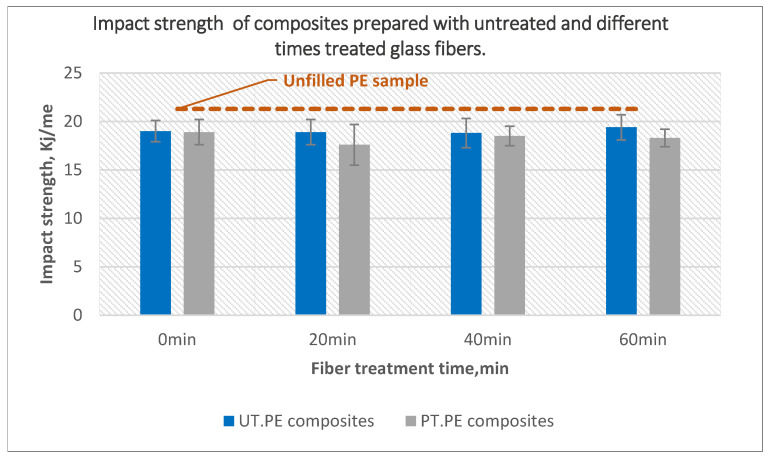
The effect of glass fiber treatment time on the impact strength of the composites prepared using untreated and 40-min treated PE with 10 wt% of untreated and plasma-treated glass fibers.

**Figure 16 polymers-14-02592-f016:**
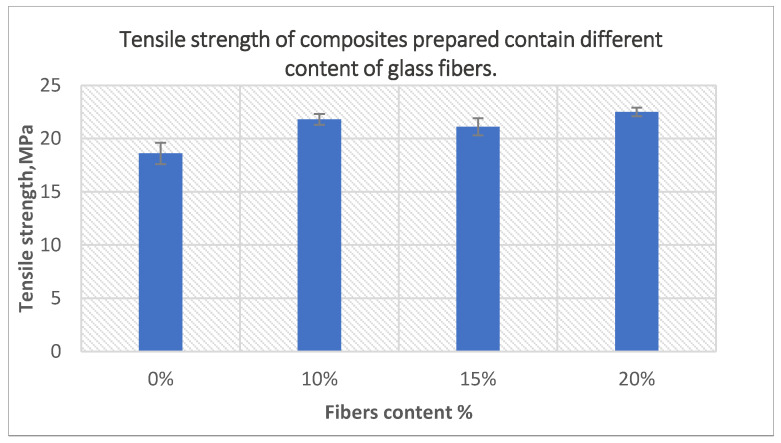
The effect of fiber content on the tensile strength of the composites prepared using 10-min treated powder and 40-min treated fibers.

**Figure 17 polymers-14-02592-f017:**
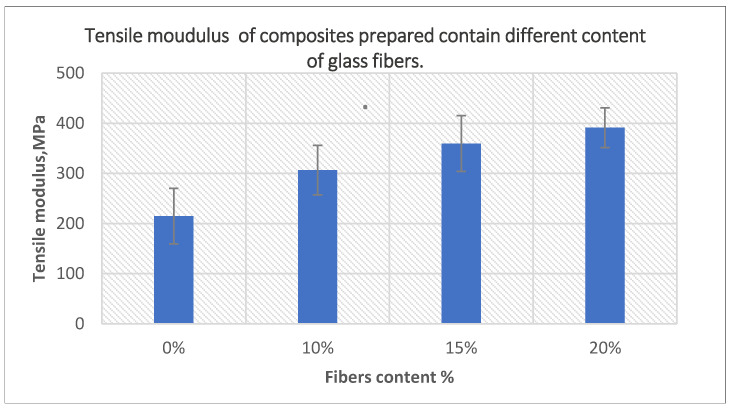
The effect of fiber content on the tensile modulus of the composites prepared using 10-min treated powder and 40-min treated fibers.

**Figure 18 polymers-14-02592-f018:**
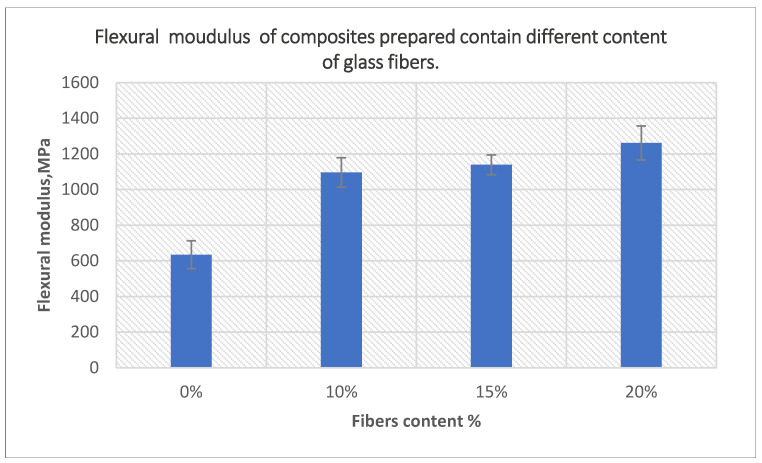
The effect of fiber content on the flexural modulus of the composites prepared using 10-min treated powder and 40-min treated fibers.

**Figure 19 polymers-14-02592-f019:**
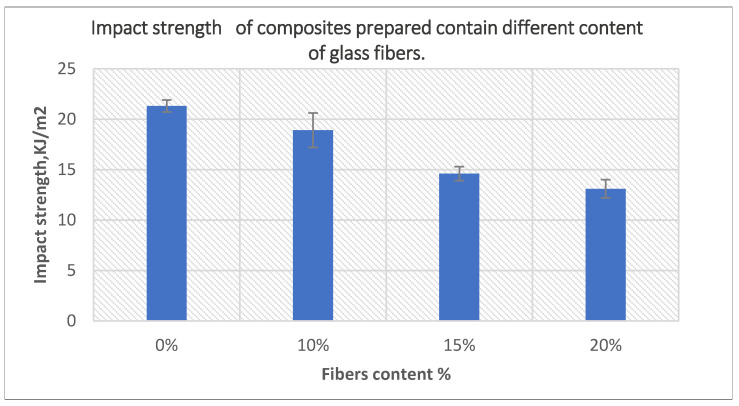
The effect of fiber content on the impact strength of the composites prepared using 10-min treated powder and 40-min treated fibers.

**Figure 20 polymers-14-02592-f020:**
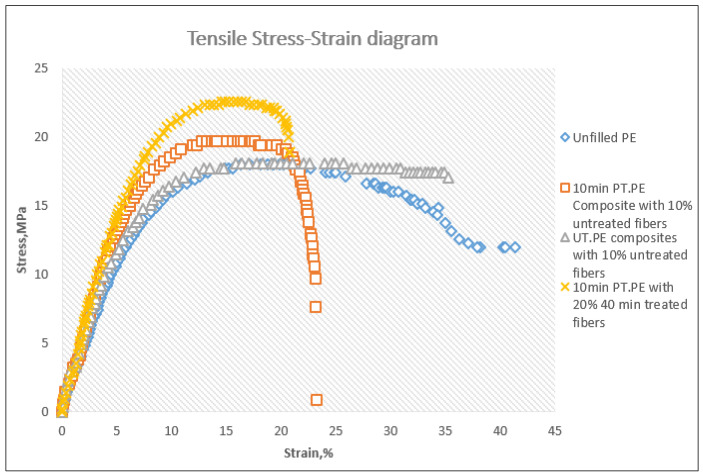
Examples of tensile stress–strain diagrams of different combinations of the rotomolded samples.

**Figure 21 polymers-14-02592-f021:**
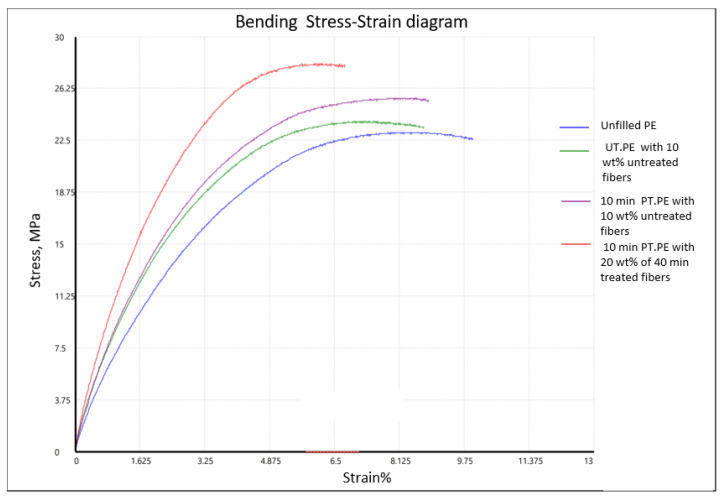
Examples of bending stress–strain diagrams of different combinations of the rotomolded samples.

**Figure 22 polymers-14-02592-f022:**
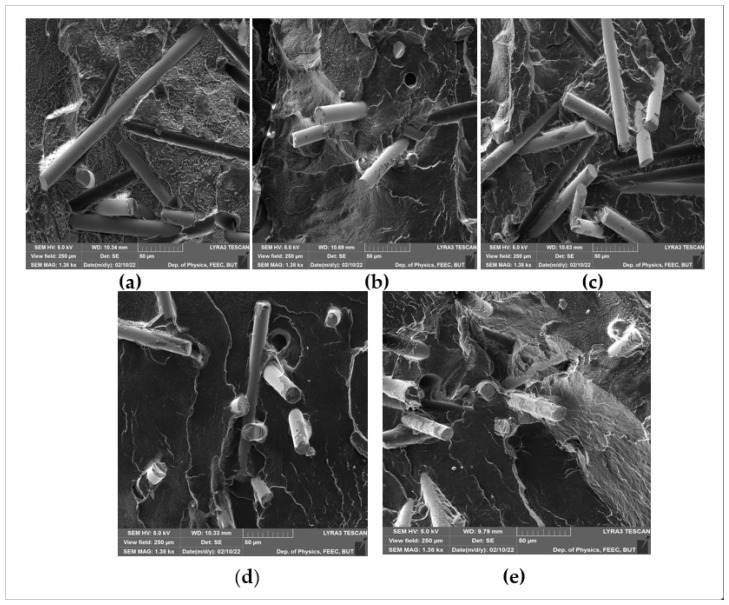
SEM images of fracture surface of composites prepred uisng untreated and plasma-treated polyethylene: (**a**) composites with untreated PE, (**b**) composites with 1-min plasma-treated PE, (**c**) composites with 3-min plasma-treated PE, (**d**) composites with 5-min plasma-treated PE, (**e**) composites with 10-min plasma-treated PE.

**Figure 23 polymers-14-02592-f023:**
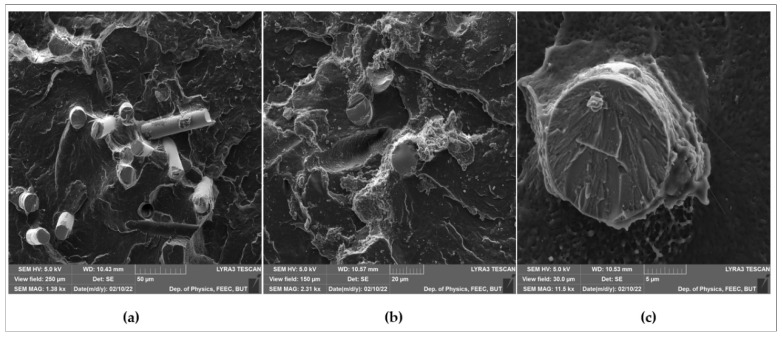
SEM images with different magnification taken from different places of the fracture surface of the composites prepared using 10-min treated powder and 40-min treated fibers: (**a**) SEM Mag: 1.3 Kx, (**b**) SEM Mag:2.31 Kx, (**c**) SEM Mag:11.5 Kx.

**Figure 24 polymers-14-02592-f024:**
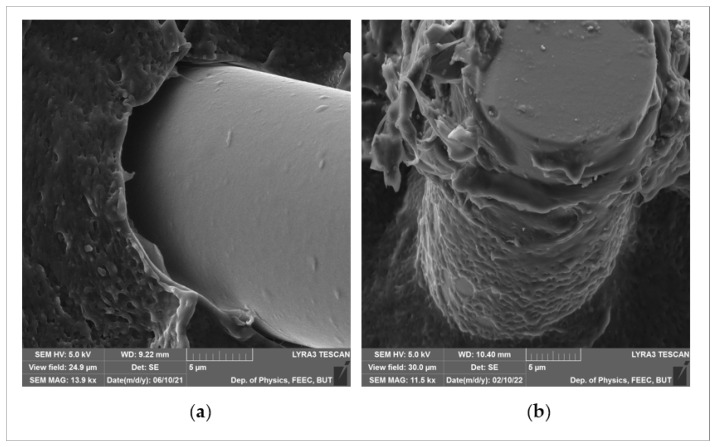
SEM images show the adhesion between PE and glass fibers: (**a**) untreated PE and untreated fiber, (**b**) 10 min plasma treated PE and 40 min plasma treated glass fibers.

## Data Availability

The data presented in this study are available on request from the corresponding author.

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
