# Peer review of "The Effect of Plasma Treatment of Polyethylene Powder and Glass Fibers on Selected Properties of Their Composites Prepared via Rotational Molding"

_polymers, 2022, doi:10.3390/polym14132592_

Round 1

Reviewer 1 Report

This is a paper containing engineering data and materials properties of polymer composites. The content could be useful for the plasma-polymer research community. However, there is a major concern in the scheme performed in this study.

Plasma treatment on powders usually faces problem of non-uniformity. This is because plasma treatment frequently treats part of the powders only. In this article, not much descriptions are made regarding the plasma treatment arrangement. Do the authors put powders in plasma machine without other special arrangement? If so, only the parts of powders faces plasma will be treated. The treatment will be non-uniform. The authors should address more in this regard.

Furthermore, I am curious about the hydrophilicity/hydrophobicity of LLDPE and glass fibers before and after plasma treatments. Do the authors have related data in this regard?

In addition, surface chemical compositions of LLDPE and glass fibers before/after plasma treatment would be interesting and related to the adhesion between LLDPE and glass fibers.

Author Response

Dear reviewer 1, 

Thank you for your valuiable comments and I here attached the answers for your comments.

Reviewer 2 Report

Dear Authors,

The manuscript submitted for review entitled "The effect of plasma treatment of polyethylene powder and glass fibers on the properties of polyethylene/glass fibers composites prepared via rotational molding" by Zoya Ghanem, Petr Špatenka, Hana Jelinek Šourkova and Sumesh K R, deals with current issues related to the influence of parameters of plasma treatment of polyethylene and glass fibers on the mechanical properties of composite products prepared via rotational molding.

The manuscript is written correctly and the presented content corresponds to the formulated topic.  The paper contains an adequate theoretical introduction, the aim and scope of the work and the experimental part ending with correctly formulated conclusions. The literature sources were selected appropriately to the subject of the work. The work meets the formal requirements, its volume is sufficient.

However, there are several issues that should be clarified:

1. In the title of the paper it should be specified what properties are meant, e.g. "selected mechanical properties", because only such properties were studied,

2. the K and R letters next to the word Sumesh should be clarified,

3. Point 1 of the manuscript "Introduction" should be shortened. For example, there is no need to discuss the principle of operation and essence of rotomolding, these are academic basics,

4. line 118, should be: selected mechanical properties.

5. in section 2.4 of the manuscript, present in tabular form the properties of the polyethylene and glass fibres tested,

6. Please specify MFI value,

7. Please give SI units e.g. kg/m3 not g/cm3, 

8. When providing information on materials and equipment, please indicate the manufacturer, city and country - this is not done in all cases, 

9. Please. explain why the authors tested only tensile strength and tensile modulus? What entitles them to state that they tested tensile properties (section 2.4)? and what about strain?

10. The word "Tenisel" in Figure 3 should be corrected,

11. Please, explain why the authors only studied the flexural modulus? What about the other properties determined in the bending test, e.g. flexural strength, deflection at fracture and modulus of elasticity?

12. In Figure 5 the unit of impact strength should be corrected to kJ/m2,

13. The manuscript should include example figures showing stress-strain relationships in tension and bending tests,

14. The paper should explain in detail how the test specimens were obtained, what was obtained by rotomolding, what the mold looked like, what walls of casting were used and how the specimens were cut out for tensile and flexural testing and impact strength,

15. In the discussion presented, you should not only explain the phenomena occurring and the results obtained, but also support the correctness of your statements with results obtained by other authors,

16. You should write MPa and not Mpa,

17. The terms UTPE and TPE are misleading because TPE stands for polyester thermoplastic elastomer - should be changed.

18. Check all the literature carefully, as several items are written incorrectly - several items lack volume, page range, etc. 

19. The manuscript should be checked again to ensure that it fully complies with the journal's guidelines.

After taking these corrections into account, the manuscript can be published in the journal "Polymers".

Best regards,

Author Response

Dear Reviewer 2,

Thank you for your valuable Please see the attahcment for the answers for your comment. 

Round 2

Reviewer 1 Report

The authors have well responded to the reviewers' comments.

Reviewer 2 Report

Dear Authors,

Thank you for your comprehensive improvement of the manuscript. The modifications made to the text are correct and sufficient, and enhance the scientific quality of the manuscript, which can be published as is.

Best regards,